# Preparation and purification of mono-ubiquitinated proteins using Avi-tagged ubiquitin

Winnie Tan[1,2], Vincent J. Murphy[1], Aude Charron[1,3], Sylvie van Twest[1], Michael Sharp[1], Angelos Constantinou[4], Michael W. Parker[5,6], Wayne Crismani[1,2], Rohan Bythell-Douglas[1,2], Andrew J. Deans[1,2]*

**1** Genome Stability Unit, St. Vincent's Institute of Medical Research, Fitzroy, Victoria, Australia, **2** Department of Medicine (St. Vincent's Health), The University of Melbourne, Victoria, Australia, **3** National Graduate School of Chemistry of Montpellier (ENSCM), Montpellier, France, **4** Institute of Human Genetics (IGH), Centre National de la Recherche Scientifique (CNRS), Université de Montpellier (UM), Montpellier, France, **5** Structural Biology Unit, St. Vincent's Institute of Medical Research, Fitzroy, Victoria, Australia, **6** Bio21 Institute, University of Melbourne, Parkville, Victoria, Australia

* adeans@svi.edu.au

**Data Availability Statement:** All relevant data are within the manuscript and its Supporting Information files. Plasmids used in this study are

## Abstract

Site-specific conjugation of ubiquitin onto a range of DNA repair proteins regulates their critical functions in the DNA damage response. Biochemical and structural characterization of these functions are limited by an absence of tools for the purification of DNA repair proteins in purely the ubiquitinated form. To overcome this barrier, we designed a ubiquitin fusion protein that is N-terminally biotinylated and can be conjugated by E3 RING ligases onto various substrates. Biotin affinity purification of modified proteins, followed by cleavage of the affinity tag leads to release of natively-mono-ubiquitinated substrates. As proof-of-principle, we applied this method to several substrates of mono-ubiquitination in the Fanconi anemia (FA)-BRCA pathway of DNA interstrand crosslink repair. These include the FANCI: FANCD2 complex, the PCNA trimer and BRCA1 modified nucleosomes. This method provides a simple approach to study the role of mono-ubiquitination in DNA repair or any other mono-ubiquitination signaling pathways.

## Introduction

Ubiquitination is the process of covalent attachment of ubiquitin to a substrate protein. Conjugation can be via a single ubiquitin, called mono-ubiquitination, or to a single ubiquitin which is then further modified by a series of ubiquitin molecules, called poly-ubiquitination [1]. Ubiquitination of substrates contributes to the regulation of all aspects of cellular function, including the DNA damage response. Coordinated ubiquitination of a substrate is achieved by an enzymatic cascade of E1, E2 and E3 enzymes. In particular several mono-ubiquitinated proteins perform critical but unknown functions within the Fanconi anemia (FA)-BRCA pathway [2–4].

A primary role of the FA-BRCA pathway is to respond to DNA interstrand crosslinks (ICLs), a type of DNA damage caused by chemotherapeutic agents which covalently link complementary strands of DNA. ICLs stall DNA replication forks and subsequently inhibit DNA

available from Addgene, with accession codes listed in Table 1.

**Funding:** This work was supported by a grant from the National Health and Medical Research Council, Australia (GNT1123100 to A.J.D. and GNT1156343 to WC), Maddie Riewoldt's Vision (SVI-MRV2017G) and the Victoria government IOS program. WT is recipient of a research training scholarship from the University of Melbourne, R. B-D is funded by the National Breast Cancer Foundation, W.C is an NHMRC Career Development Fellow (GNT1129757) and Maddie Riewoldt's Vision Fellow (WC-MRV2016) and AJD is a Victorian Cancer Agency mid-career fellow. The funders had no role in study design, data collection and analysis, decision to publish, or preparation of the manuscript.

**Competing interests:** The authors have declared that no competing interests exist.

transcription and replication [5]. The removal of ICLs by the FA-BRCA pathway is therefore crucial to genome stability. There are at least three essential mono-ubiquitination events within the FA-BRCA pathway. First, replication stalling induces mono-ubiquitination of PCNA, a homo-trimeric substrate for the E2 conjugating enzyme RAD6 and E3 ligase RAD18 [6]. The mono-ubiquitination of PCNA at lysine 164 site recruits the DNA repair proteins REV1 and Pol η to the DNA damage site [7]. Second, PCNA-Ub is necessary for the recruitment of the FA "core complex" of proteins consisting of FANCA, FANCB, FANCC, FANCE, FANCF, FAAP20, FAAP100 and the E3-RING ligase FANCL [8]. FANCL within the FA core complex is responsible for the coordinated conjugation of ubiquitin to both FANCD2 and FANCI within the FANCI:FANCD2 complex [9–11]. The dually-mono-ubiquitinated FANCI:FANCD2 complex has been hypothesized to recruit the nucleases such as SNM1A and FAN1 that are involved in the first incision of ICL called unhooking [12–14]. Third, unhooking of the ICL creates a free DNA end that stimulates nucleosome mono-ubiquitination by BRCA1, which stimulates recombination and replication fork stabilization [15]. BRCA1, and its partner protein BARD1, interact through a RING-domain heterodimer to mono-ubiquitinate nucleosomes in concert with the UBCH5C E2-enzyme [4, 16]. The BRCA1:BARD1 E3-complex specifically directs mono-ubiquitination to lysines 125, 127 and 129 of histone H2A both *in vitro* and *in vivo* [4], which is hypothesized to promote long-range resection of the DNA by SMARCAD-mediated chromatin unloading [17].

The specific details of direct protein interactions and events mediated by mono-ubiquitination described above are poorly understood at the mechanistic level. This is because of the difficulty in purifying sufficient quantities of purely ubiquitinated material for biochemical analysis. Several strategies have previously been tried. For example, mono-ubiquitinated PCNA has been made using only a modified E2 enzyme, UBCH5C (S22R) [18, 19]. Ubiquitinated PCNA-Ub has also been generated using non-enzymatic approaches through chemical ligation where the lysine acceptor residue has been substituted with a chemically-reactive variant [20, 21]. For histones, which are very small proteins, intein-mediated ligation of histone fragments and ubiquitin-conjugated peptides were used successfully [22], although this method has not been described for K127 or K129-modified H2A. Generation of mono-ubiquitinated FANCI and FANCD2, which are an order of magnitude larger than PCNA and histones has proven more challenging. One approach generated mono-ubiquitinated FANCI and mono-ubiquitinated FANCD2 monomers separately, using a hyperactive UBE2T-E2 enzyme and truncated FANCL ligase [23, 24]. However, the substrate of the FA core complex is DNA-bound FANCI:FANCD2 complex, and ubiquitination of isolated subunits followed by mixing may not recapitulate the *in vivo* properties of FANCI:FANCD2 complex. We and others have shown that efficient *in vitro* mono-ubiquitination of FANCI:FANCD2 complex requires recombinant FA E3-ligase complex (FANCB-FANCL-FAAP100 (BL100)), adaptor complex (FANCC-FANCE-FANCF (CEF)) and DNA [11, 25].

Here we describe the purification of natively mono-ubiquitinated FANCI:FANCD2 complex, along with other mono-ubiquitinated components of the FA-BRCA pathway. This procedure makes use of a modified ubiquitin, which is fused to the Avi-tag, a peptide sequence that is efficiently biotinylated in a site-specific manner in *E. coli* by the BirA biotin-ligase [26]. Using *in vitro* reconstituted E1-E2-E3 reactions that recapitulate the endogenous mono-ubiquitination reactions, we can then isolate solely ubiquitinated proteins by biotin affinity purification and elution by proteolytic cleavage of the Avi tag. While the examples we provide are all within the FA-BRCA pathway, this procedure provides reagents for biochemical and biophysical analysis of any protein where *in vitro* ubiquitination can be reconstituted.

# Materials and methods

## Protein purification

Table 1 outlines the plasmids used in this study. Plasmids were propagated using NEB-10-beta competent cells (NEB) and purified using Monarch miniprep kits (NEB). Bacmids were generated using the Multibac system [27] and purified using alkaline lysis method followed by isopropanol precipitation and resuspension in TE buffer.

## Avidin-ubiquitin

We designed a ubiquitin coding construct encoding an N-terminal 10xHis tag, followed by a biotinylatable AviTag, a 3C protease site and then ubiquitin at the C-terminus (Fig 1A). This construct is hence referred to as Avi-ubiquitin. Avi-ubiquitin was biotinylated by co-expression with BirA biotin ligase *in vivo* in *E. coli* in 500 mL auto-induction media [30] supplemented with 11 mg biotin (Sigma-Aldrich) overnight at 37 ˚C. Cells were harvested by centrifugation at 6000 x *g* for 15 min at 4 ˚C. The cell pellet was resuspended in buffer A (1x PBS, 20 mM imidazole, 10% (w/v) glycerol and 1 mM DTT) supplemented with 1 mg/mL lysozyme (Sigma-Aldrich), 1x bacterial protease inhibitor (Sigma-Aldrich), and lysed by sonication on ice. The lysate was clarified by centrifugation at 16,000 x *g* for 40 min at 4 ˚C. The supernatant was mixed with Ni-NTA resin (Thermo Fisher) pre-equilibrated in buffer A for 2 hrs at 4 ˚C, followed by a 10 column volume (CV) wash with buffer A. Avi-ubiquitin was eluted with a linear gradient of buffer A from 20 mM to 500 mM imidazole over 16 CV. Fractions containing Avi-ubiquitin were pooled and dialyzed against buffer B (1xPBS, 1 mM DTT) using SnakeSkin dialysis tubing with 3.5 kDa molecular weight cutoff (MWCO) (Thermo Fisher). Dialyzed eluates were then loaded onto a Vydac 1x25 cm C18 reverse phase column (VWR) with buffer B containing 0.1% trifluoroacetic acid (TFA). Elution was conducted over a linear gradient to 0.1% TFA, 70% acetonitrile in buffer B. The fractions containing Avi-

**Table 1. Plasmids used in this study.**

| Plasmid | Protein | Selection | Reference | Use |
|---|---|---|---|---|
| Fastbac1-FLAG-xFANCI | *X.laevis* FANCI | Amp, Gent | [11] | a |
| pFastbac1-StrepII-xFANCD2 | *X.laevis* FANCD2 | Amp, Gent | [11] | a |
| pFastbac1-FLAG-hFANCD2$^{opt}$ | Codon optimized human FANCD2 | Amp, Gent | gene synthesis by Gene Art (Thermo Fisher). Deposited Addgene: #134904 | a |
| pFL-EGFP-His-hFANCI | Human FANCI | Amp, Gent | [11, 25] | a |
| pFL/pSPL-EGFP-FLAG-B-L-100 | FANCL, FAAP100 and codon optimized FANCB | Amp, Gent, Spec | [11, 25] | a |
| pFL-MBP-C-E-F | FANCC, FANCE, FANCF | Amp, Gent | [11, 25] | a |
| pGEX-KG-GST-UBE2T | Codon optimized human UBE2T | Amp | [11, 25] | b |
| pet16b-Avi-ubiquitin_rbs_BirA | Ubiquitin, *E. coli* biotin ligase (operon) | Amp | gene synthesis by Gene Universal. Deposited Addgene: #134897 | b |
| pET16b-PCNA | PCNA | Amp | gene synthesis by Gene Universal. Deposited Addgene: #134898 | b |
| pFastbac1-FLAG-BRCA1$^{Δexon11}$ | Codon optimized human BRCA1$^{Δexon11}$ | Amp, Gent | gene synthesis by Gene Universal. Deposited Addgene: 137167 | a |
| pFastbac1-StrepII-BARD1 | Codon optimized human BARD1 | Amp, Gent | gene synthesis by Gene Universal. Deposited Addgene: 137166 | a |
| pET28a-UBCH5C$^{S22R}$ | UBCH5C | Amp | a gift from Rachel Klevit [28] (Addgene plasmid # 12644) | b |
| pSRK2706-GST-HRV-3C protease | HRV-3C protease | Amp | A gift from David Waugh [29](Addgene plasmid # 78571) | b |

Amp = Ampicillin, Gent = Gentamycin, Spec = Spectinomycin, use a = bacmid generation for baculovirus expression and protein purification, use b = *E. coli* BL21 transformation for protein expression and purification

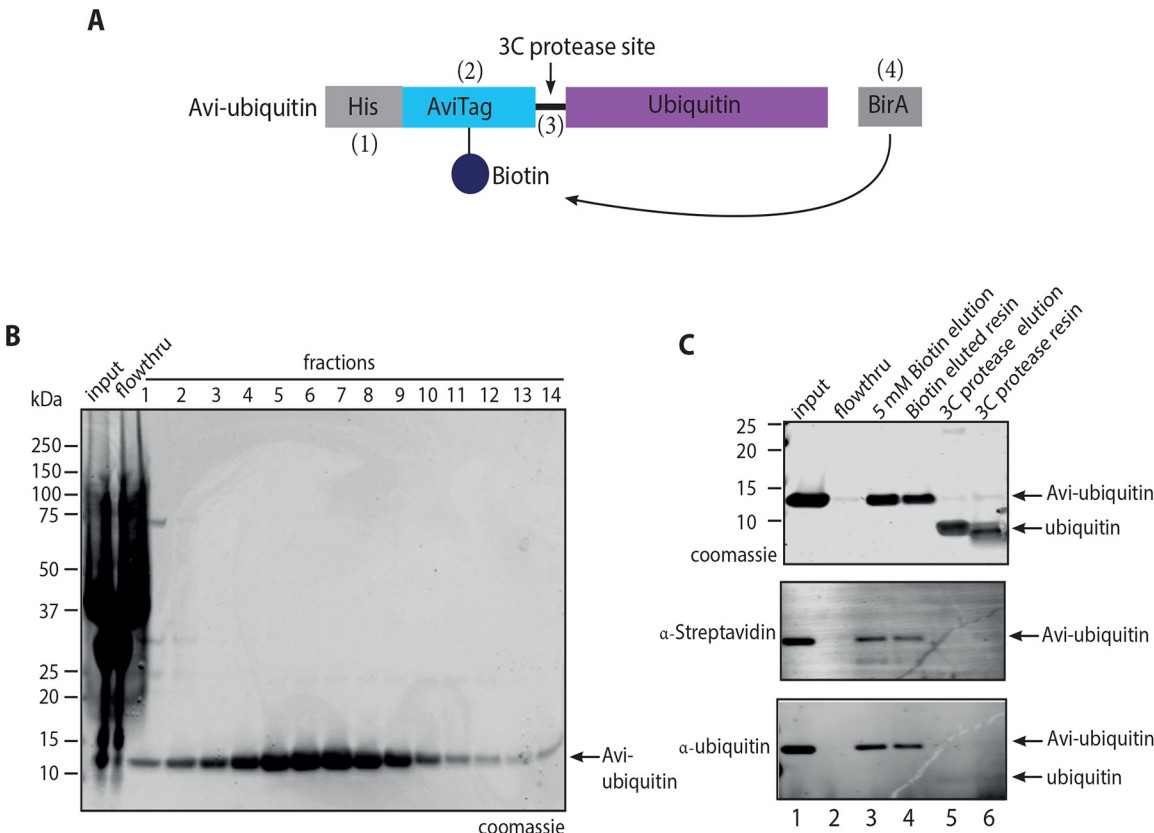

**Fig 1. Purification of recombinant Avi-ubiquitin. (A)** Schematic of the Avi-ubiquitin construct used for expression and purification of recombinant Avi-ubiquitin from E. coli cells. #1–4 refers to features of Avi-ubiquitin required for biotinylation and purification. **(B)** Coomassie stained SDS-PAGE gel showing purification of recombinant Avi-ubiquitin using Ni-NTA resin. **(C)** Coomassie stained SDS-PAGE gel (top), anti-Streptavidin (middle) and anti-ubiquitin (bottom) western blots showing elution of Avi-ubiquitin by using 5 mM biotin or by cleavage of bound ubiquitin using 3C protease. Experiments in **(B-C)** were performed in triplicate with similar results. Note: cleaved ubiquitin is below the 10 kDa lower limit for optimal binding to PVDF membranes, so only weakly observed by western blot.

ubiquitin were then lyophilized in a freeze dryer, and resuspended in 1xPBS to a final concentration of 2 mg/mL. This protein was then aliquoted prior to plunge freezing in liquid nitrogen and storage at -80 ˚C. To verify that Avi-ubiquitin was biotinylated, purified Avi-ubiquitin was gently mixed with SoftLink™ monomeric avidin resin (Promega) pre-equilibrated in buffer B for 1 hr at 4 ˚C. The resin was washed with 10 CV buffer B and Avi-ubiquitin was eluted with buffer B supplemented with 5 mM Biotin (ThermoFisher) or 100 nM HRV-3C protease.

## FANCI:FANCD2

For expression of the FANCI:FANCD2 complex, His-FANCI and FLAG-FANCD2 plasmids were integrated into the MultiBac expression system and Bacmid constructs generated as described [27]. Purified bacmids were transfected into *Spodoptera frugiperda* (Sf9) cells in SF900-II medium using FuGene HD transfection reagent (Promega). Infection was monitored by GFP co-expression. A third-generation amplification of the baculovirus was used to infect *Trichoplusia ni* (High Five™, Life Technologies) cells for protein production. 200 mL High Five insect cells at $1.0 \times 10^6$ cells/mL were co-infected with Multibac baculoviruses expressing His-FANCI and FLAG-FANCD2. At 72 hours post-infection, cells were harvested by

centrifugation at 500 x $g$ for 5 min at 4 ˚C. The harvested cells were resuspended in buffer C (20 mM Tris-HCl pH 8.0, 100 mM NaCl, 10% glycerol) supplemented with 1 mM EDTA and 1X mammalian protease inhibitor (Sigma-Aldrich) and lysed by sonication on ice. The lysate was clarified by centrifugation at 16,000 x $g$ for 45 min at 4 ˚C. The supernatant was added to M2 anti-FLAG affinity resin (Sigma-Aldrich) pre-equilibrated in buffer C, and gently mixed at 4˚C for 2 hrs. The resin was washed with 10 CV of buffer C and eluted in the buffer C supplemented with 5 μg/mL FLAG peptide. Fractions containing FANCI:FANCD2 were pooled and loaded on to TALON metal resin (Takara Bio) pre-equilibrated in buffer C and eluted with buffer C supplemented with 250 mM imidazole. FANCI:FANCD2 complex containing fractions were pooled, aliquoted, plunge-cooled in liquid nitrogen and stored at -80 ˚C.

*Xenopus laevis* FANCI:FANCD2, human UBE2T, Flag-FANCB-FANCL-FAAP100 and MBP-FANCC-FANCE-FANCF were purified as previously described [11]. His-UBE1 was purchased from Boston Biochem.

## PCNA

Human PCNA was purified as described [31] with slight modifications. Untagged PCNA was overexpressed in BL21(DE3) cells in autoinduction media at 37 ˚C overnight. Cells were harvested at 6000 x $g$ for 15 min at 4 ˚C, resuspended in 50 mM Tris-HCl pH 8.0, 0.1 M NaCl, 10% glycerol, 1 mM EDTA and 1X bacterial protease inhibitor (Sigma-Aldrich)) and lysed by sonication on ice. After centrifugation at 16,000 x $g$ for 30 min at 4 ˚C, the clarified lysate was applied to Q Sepharose Fast Flow column (GE Healthcare) pre-equilibrated in buffer D (20 mM Tris-HCl pH 8.0, 10% glycerol). The loaded column was washed with 15 CV of 27% buffer D (buffer C + 1 M NaCl), followed by a 2.7 CV gradient from 27% to 54% buffer D and then a 5 CV gradient from 54% to 57% buffer D. PCNA proteins eluted in the final gradient and these fractions were assessed by SDS-PAGE. Fractions containing PCNA were dialyzed into buffer C and loaded onto HiPrep Heparin Fast Flow 16/10 (GE Healthcare), pre-equilibrated in buffer C. The loaded resin was washed with 5 CV of buffer C, followed by a 5 CV elution gradient from 0% to 100% buffer D. Eluted proteins were collected in 2 mL fractions and analyzed by SDS-PAGE. PCNA containing fractions were pooled, concentrated and buffer exchanged into buffer C using Amicon Ultra 10 kDa MWCO centrifugal filter (Millipore). The purified PCNA proteins were aliquoted, plunge-cooled in liquid nitrogen and stored at -80 ˚C in aliquots.

## BRCA1$^{\Delta exon\ 11}$:BARD1 complex

Sf9 insect cells at 1.0 x $10^6$ cells/mL were co-infected with Multibac baculoviruses expressing FLAG-BRCA1$^{\Delta exon\ 11}$ and StrepII-BARD1 for 48 hrs. Cells were harvested by centrifugation at 500 x $g$ for 5 min at 4 ˚C. The cell pellet was resuspended in buffer E (50 mM TEA pH 7.4, 150 mM NaCl, 10% glycerol) supplemented with 1 mM EDTA and 1X mammalian protease inhibitor (Sigma-Aldrich) prior to lysis by sonication. The lysate was clarified by centrifugation at 16,000 x $g$ at 4 ˚C for 30 min. The clarified lysate was gently mixed with Streptactin resin (GE Healthcare) pre-equilibrated in buffer E for 2 hrs at 4 ˚C. The loaded resin was washed in batch with 20 CV buffer E five times for 5 min each wash, and the protein was eluted in buffer E containing 0.5 mg/mL desthiobiotin (Sigma-Aldrich). The purified proteins were aliquoted and plunge-cooled in liquid nitrogen prior to storage at -80 ˚C.

## Nucleosome reconstitution

The DNA used to generate nucleosomes was the 147 bp Widom 601 sequence [32] prepared by annealing complementary single stranded 147 bp Ultramers™ (Integrated DNA

Technologies) with the sequence 5′- `ATCGAGAATCCCGGTGCCGAGGCCGCTCAATTGGTC GTAGACAGCTCTAGCACCGCTTAAACGCACGTACGCGCTGTCCCCGCGTTTTAACCGCCAA GGGGATTACTCCCTAGTCTCCAGGCACGTGTCAGATATATACATCCGAT`–3′. To prepare nucleosomes, 10 μM recombinant human histone octamers (histonesource.com) were buffer exchanged into buffer F (2 M NaCl, 10 mM Tris-HCl pH 7.5, 1 mM EDTA, 5 mM DTT) using a 7 kDa MWCO Zeba desalting column (Thermo Fisher), combined at a 1:1 molar ratio with Widom 601 dsDNA in buffer F and incubated for 30 min at 4 ˚C. The reconstitution reaction was dialyzed in buffer F with decreasing salt concentration stepwise from 2.0 M, 1.0 M, 0.8 M, 0.6 M to 0.2 M NaCl with incubation time of 2–3 hrs with 0.8 M NaCl buffer dialysis being an overnight step using a 7 kDa MWCO Slide-A-Lyzer mini dialysis unit (Life Technologies). The nucleosome reconstitution was confirmed by electromobility shift assay using 6% native acrylamide gel, visualized with ethidium bromide and Coomassie blue. The reconstituted nucleosomes were stored at 4 ˚C and used within 4 weeks of production.

## UBCH5C (S22R)

UBCH5C (S22R) purification was performed as described previously [28].

## HRV-3C protease

GST-tagged HRV-3C protease was overexpressed in BL21 (DE3) in auto-induction media at 37 ˚C overnight. Cells were harvested by centrifugation at 6000 x $g$ for 15 min at 4 ˚C. The cell pellet was resuspended in buffer G (50 mM Tris-HCl pH 7.4, 100 mM NaCl, 1 mM EDTA, 10% glycerol and 1 mM DTT), supplemented with 1 mg/mL lysozyme (Sigma-Aldrich) and 1x bacterial protease inhibitor (Sigma-Aldrich), and lysed by sonication on ice. The lysate was clarified by centrifugation at 16,000 x $g$ at 4 ˚C for 30 min. The supernatant was gently mixed with pre-equilibrated GSH resin for 2 hrs at 4 ˚C, washed with 10 CV buffer G and eluted with buffer G supplemented with 20 mM reduced glutathione. Fractions containing GST-HRV3C were pooled, aliquoted, plunge-cooled in liquid nitrogen and stored at -80 ˚C. Note: HRV-3C protease can also be purchased from commercial sources as PreScission® Protease.

## *In vitro* ubiquitination assays

**FANCI:FANCD2 complex.** Ubiquitination reactions contained 100 nM purified human or *Xenopus* FANCI:FANCD2, 50 nM UBE1 (Boston Biochem), 100 nM UBE2T, 100 nM BL100, 100 nM CEF, 7.5 ng/μl pUC19 plasmid DNA, 10 μM Avi-ubiquitin and 2 mM ATP in 50 mM Tris-HCl pH 7.4, 150 mM NaCl, 2.5 mM $MgCl_2$, 0.01% Triton X-100 and 1 mM DTT. Reactions were incubated for 90 min at 37 ˚C.

**PCNA.** Ubiquitination reactions contained 50 nM UBE1, 100 nM UBCH5C, 100 nM PCNA, 10 μM Avi-ubiquitin and 2 mM ATP in 50 mM MMT buffer (DL-Malic acid, MES monohydrate, Tris) pH 9.0, 25 mM NaCl, 2.5 mM $MgCl_2$, 0.01% Triton X-100 and 1 mM DTT. Reactions were incubated for 90 min at 37 ˚C.

**Nucleosomes.** Ubiquitination reactions contained 50 nM UBE1, 150 nM UBCH5C, 100 nM nucleosomes, 100 nM BRCA1:BARD1 complex, 10 μM Avi-ubiquitin and 2 mM ATP in 50 mM Tris-HCl pH 7.4, 25 mM NaCl, 2.5 mM $MgCl_2$, 0.01% Triton X-100 and 1 mM DTT. Reactions were incubated for 90 min at 37 ˚C.

## Purification of mono-ubiquitinated proteins

At the completion of reactions, ubiquitination mixtures were added to Avidin agarose resin (Pierce) and incubated with gentle mixing for 2 hrs at 4˚C. Non-ubiquitinated proteins were

washed away from the resin with 10 CV of buffer H (50 mM Tris-HCl pH 7.4, 150 mM NaCl, 10% glycerol and 1 mM DTT). Bound, ubiquitinated proteins were cleaved overnight by 100 nM GST-HRV-3C in buffer H. The GST-HRV-3C protease was removed using glutathione affinity purification and the purified, ubiquitinated complexes were dialyzed into buffer H using a 20 kDa MWCO dialysis membrane (Pacific Laboratory) to remove unconjugated ubiquitin.

### Antibodies

The following antibodies were used in this study: rabbit polyclonal antibodies against PCNA (ab15497, Abcam), histone H2A (ab18255, Abcam) and StrepII (ab76949, Abcam); mouse monoclonal antibodies against FLAG (Sigma, M2) and ubiquitin (ab7254, Abcam); and DyLight™ 800-conjugated Streptavidin (S000-32, Rockland).

### Mass spectrometry

Protein bands were destained and dehydrated with 500 μL acetonitrile and subsequently reduced with 500 μL 10 mM dithiothreitol (DTT) in 25 mM ammonium bicarbonate ($NH_4HCO_3$) at 55 ˚C for 1 hr and then alkylated with 200 μL 55 mM iodoacetamide in 25 mM $NH_4HCO_3$ at room temperature for 45 min in the dark. Samples were digested overnight with 125 ng trypsin (Promega) per band at 37 ˚C in a final volume of 20 μL. The resultant peptides were subjected to sonication with 50 μL of 50% acetonitrile, 5% formic acid and prepared for mass spectrometry by concentration under vacuum to a 10–15 μL final volume. Peptide mixtures were analyzed by reverse-phase HPLC-MS/MS. Mass spectra of digested protein gel bands were obtained using an electrospray ionisation (ESI)-quadrupole-time-of-flight Q Exactive Plus mass spectrometer (ThermoFisher Scientific). The analysis program MASCOT was used to identify mono-ubiquitination sites on FANCI:FANCD2, PCNA, BRCA1$^{\Delta exon11}$:BARD1 and nucleosomes.

## Results

### Expression, purification and quality control of recombinant proteins

In order to purify natively mono-ubiquitinated proteins, we engineered a bacterial expression construct that encodes human ubiquitin with: 1) an N-terminal 10xHis tag for affinity purification, 2) an Avi-tag for site-specific biotinylation [33] and biotin-affinity purification, 3) a 3C protease site for site-specific protease cleavage, leaving native ubiquitin with one artefactual N-terminal proline,z and 4) a second ORF for co-expression of BirA biotin ligase for efficient bacterial biotinylation of recombinant protein [26] (Fig 1A). From this construct, we obtained 21.3 mg of Ni-affinity purified recombinant Avi-ubiquitin from 500 mL *E. coli* culture (Fig 1B). To verify that the Avi-ubiquitin purified by this method was biotinylated, we loaded the purified recombinant Avi-ubiquitin onto SoftLink™ monomeric avidin resin (Promega), which binds only the biotinylated form, but permits elution of bound proteins using biotin (~50% Avi-ubiquitin were eluted, lanes 1–3, Fig 1C). We observed that there was only a very faint band (<5%) corresponding to Avi-ubiquitin in the flow-through (lane 2, Fig 1C), demonstrating the efficient (>95%) biotinylation of the Avi-ubiquitin. Additionally, treatment of the bound sample with HRV-3C protease resulted in release of >95% of the cleaved ubiquitin while the Avi-tag remained immobilized (lanes 4–6, Fig 1C).

### Avi-ubiquitin can be used to efficiently mono-ubiquitinate FANCI: FANCD2, PCNA and nucleosomes

We have previously shown that the maximal mono-ubiquitination of *Xenopus laevis* FANCI: FANCD2 complex requires the FA RING-E3 ligase subcomplex: FANCB-FANCL-FAAP100

(BL100), and the adaptor protein: FANCC-FANCE-FANCF (CEF) [11]. We have previously preferred the *X. laevis* xFANCI:xFANCD2 complex for mono-ubiquitination reactions because the native human *FANCD2* cDNA sequence is highly unstable during bacterial propagation and baculovirus generation. Here, we overcame the problem of cDNA instability, by re-engineering the protein coding sequence via codon optimization and gene synthesis. This new synthetic construct was stable after repeated propagation and used for baculovirus generation. We obtained 1 mg of pure recombinant human FANCI:FANCD2 complex from 200 mL Hi5 insect cell expression culture (S1A Fig). The two human proteins are closer in size than the corresponding *X. laevis* proteins, when separated on a 3–8% Tris-acetate gel (S1B Fig).

To determine if the same factors are also required to mono-ubiquitinate human and *X. laevis* FANCI:FANCD2 complexes, we set out to purify human FANCI:FANCD2 complex for use in ubiquitination assays (Fig 2A, **lane 2**, Note: human FANCI:FANCD2 runs closer to each other on 4–12% Bis-Tris gel when run in 1X MES buffer). Both human and *X. laevis* FANCI:FANCD2 complex can be mono-ubiquitinated at 25 °C, however the *X. laevis* FANCI: FANCD2 complex was more efficiently ubiquitinated at this temperature (S1C Fig). After testing at various temperatures, we found human FANCI:FANCD2 complex, is most efficiently mono-ubiquitinated *in vitro* at 37 °C (Fig 2B).

Mono-ubiquitination of PCNA substrate by the modified E2 enzyme UBCH5C (S22R) has been well described [18, 19]. As per the optimized *in vitro* ubiquitination conditions of [18] we were able to achieve ~50% mono-ubiquitinated PCNA with Avi-ubiquitin, in a low salt and high pH buffer (50 mM MMT pH 9.0, 25 mM NaCl, 2.5 mM $MgCl_2$, 0.01% Triton X-100 and 1 mM DTT) (Fig 2C).

For the *in vitro* mono-ubiquitination of nucleosomes, we purified human BRCA1:BARD1 complex with a yield of approximately 0.5 mg pure protein per 200 mL of Sf9 insect cell culture (Fig 2A, **lane 3**). The form of BRCA1 used was a splice variant which lacks exon 11 (BRCA1$^{\Delta exon11}$). This variant of BRCA1 has the following properties: (a) it is highly expressed in some mammary cell lines [34], (b) it contains the conserved N-terminal RING E3 ligase domain and C-terminal BRCT repeats critical for BRCA1 function in DNA repair and breast cancer suppression [35] and (c) is able to rescue the otherwise deleterious deletion of BRCA1 during mouse development [36]. Consistent with published reports [37], we show that BRCA1$^{\Delta exon11}$:BARD1 auto-ubiquitinates on multiple lysine residues in the presence of UBCH5C E2 enzyme (Fig 2D). Mass spectrometry analysis confirmed that BRCA1 and BARD1 were auto-ubiquitinated at multiple lysine sites (S3D Fig). Importantly, we also show that our recombinant BRCA1$^{\Delta exon11}$:BARD1 protein can mono-ubiquitinate histone H2A within a nucleosome (Fig 2E). Within 30 min, ~40% mono-ubiquitination of histone H2A occurred using BRCA1$^{\Delta exon11}$:BARD1 (Fig 2E), and some higher products corresponding to two or three mono-ubiquitination events were also observed. This is expected, because BRCA1:BARD1 can ligate ubiquitin to lysine residues K125, K127 and K129 in histone H2A [4, 38]. Western blotting of the native PAGE separated nucleosome reacted with anti-Streptavidin and confirmed ubiquitination occurred in the context of a nucleosome (S2A–S2C Fig).

## Purification of mono-ubiquitinated FANCI:FANCD2, PCNA and nucleosomes

Mono-ubiquitination of the FANCI:FANCD2 complex, PCNA and histone H2A within nucleosomes are key regulation steps within the FA-BRCA pathway. To purify mono-ubiquitinated FANCI:FANCD2 complex, we set up *in vitro* ubiquitination reactions in a total volume of 500 µl containing Avi-ubiquitin, human FANCI:FANCD2, FA core complex components (BL100 and CEF) and plasmid DNA. Biotin affinity purification and elution of ubiquitinated

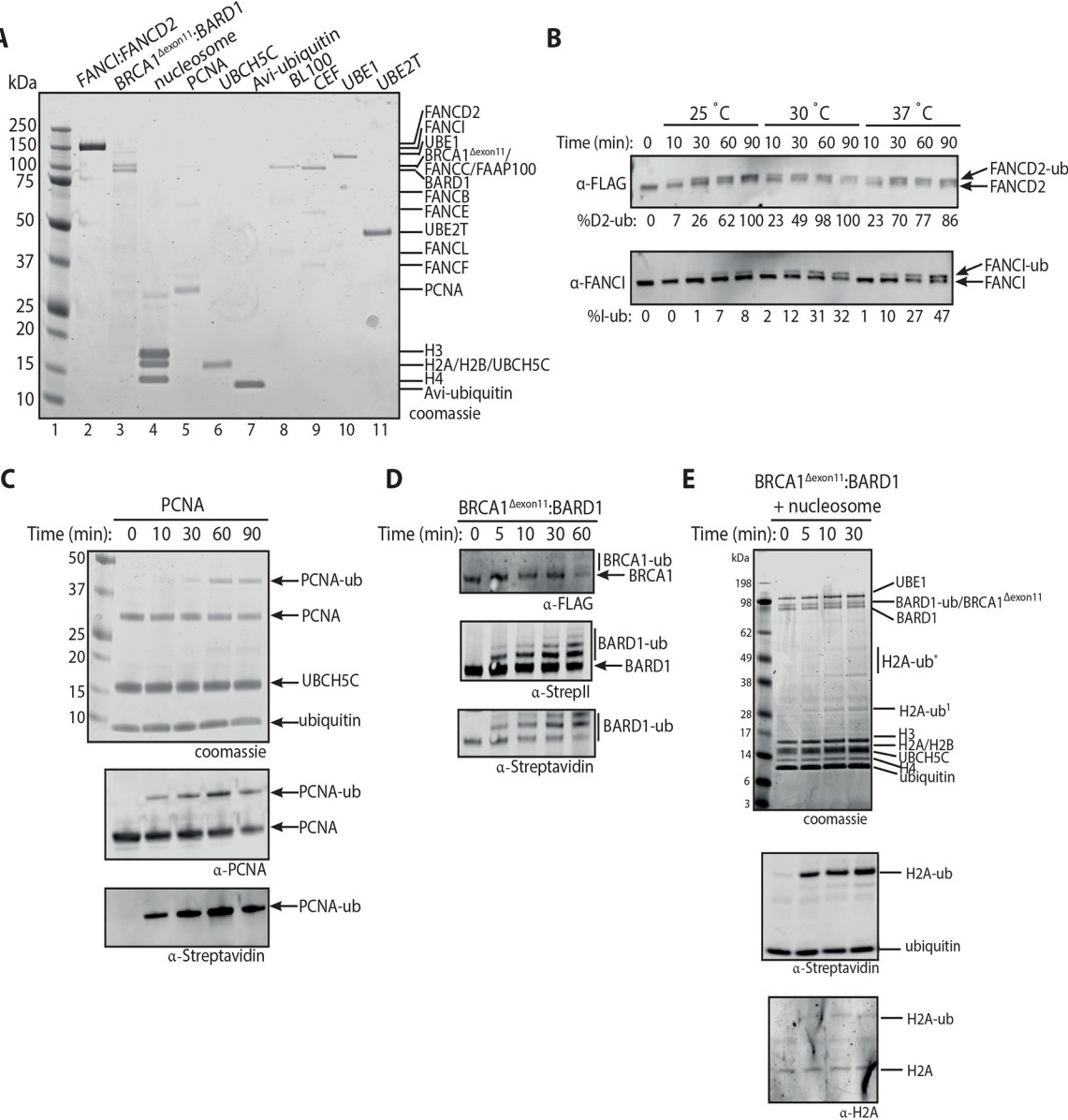

**Fig 2. Reconstitution of human FANCI:FANCD2, PCNA and nucleosome *in vitro* ubiquitination assay using Avi-ubiquitin. (A)** Coomassie stained 4–12% Bis-Tris gel run in 1X MES buffer showing purified human FANCI:FANCD2, BRCA1$^{\Delta exon11}$:BARD1, reconstituted nucleosome, PCNA, UBCH5C (S22R), Avi-ubiquitin, FANCB-FANCL-FAAP100 (BL100), FANCC-FANCE-FANCF (CEF), UBE1 and UBE2T (lanes 2–11). **(B)** Western blot of the time course ubiquitination reaction of human FANCI:FANCD2 at 25 ˚C, 30 ˚C and 37 ˚C. The percentage of mono-ubiquitinated FANCD2 or FANCI were calculated and showed under each western blot panel. **(C)** PCNA mono-ubiquitination time course experiments using UBCH5C as an E2 enzyme. **(D)** Western blots of time course experiments revealing that BRCA1$^{\Delta exon11}$ and BARD1 were auto-ubiquitinated at multiple lysine residues in the presence of UBCH5C. **(E)** Mono-ubiquitination timecourse of nucleosomes as substrates in the presence of BRCA1$^{\Delta exon11}$:BARD1 and UBCH5C. Experiments in **(B-E)** were performed in triplicate with similar results.

proteins using HRV-3C protease enzyme resulted in elution of ubiquitin (Fig 3A) and dually-mono-ubiquitinated human FANCI:FANCD2 complex (Fig 3B). Mass spectrometry analysis confirmed that FANCI and FANCD2 were mono-ubiquitinated on their native target residues, Lysine 523 and Lysine 561 respectively (S3A and S3B Fig). The purified dually-mono-ubiquitinated FANCI:FANCD2 eluted from the avidin resin is in a 1:1 stoichiometry. This observation

suggests that ubiquitination does not break apart the FANCI:FANCD2 complex, despite the ubiquitination sites being buried inside the mouse structure of non-ubiquitinated FANCI: FANCD2 [39].

PCNA mono-ubiquitination has been shown to stimulate the FA-BRCA pathway [40], however the intersection of PCNA ubiquitination and FANCD2 ubiquitination has only been investigated with FANCL as the E3 RING ligase for FANCD2 rather than the FA core complex [41]. Furthermore, in the previous study, only ~20% of PCNA molecules were mono-ubiquitinated by the cognate E3 ligase RAD18 [41]. We used Avi-ubiquitin to mono-ubiquiti-nate PCNA using an optimized reaction with the E2 UBCH5C [18]. Using this approach, we were able to generate and purify mono-ubiquitinated PCNA (Fig 3C), demonstrating that the Avi-ubiquitin system is a generalizing method for the preparation of pure mono-ubiquitinated proteins. Mass spectrometry analysis confirmed that PCNA was mono-ubiquitinated on the native target residue, Lysine 164 (S3C Fig). Protein eluted from the resin was in a ~80%-ubi-quitinated state, suggesting that in our system, mono-ubiquitination of the PCNA trimer occurs essentially simultaneously on each of the 3 subunits.

Finally, mono-ubiquitination of nucleosomal histone H2A by BRCA1:BARD1, has also been shown to regulate DNA repair in the FA:BRCA pathway [4, 16]. Using the Avi-ubiquitin system, we were able to purify mono-ubiquitinated histone H2A within a nucleosome (Fig 3D). This result demonstrates that the Avi-ubiquitin system can be used to purify natively mono-ubiquitinated nucleosomes. Not all of the ubiquitinated nucleosome was captured in these experiments, perhaps because the N-terminus of ubiquitin becomes buried with the nucleosome. Structural investigation of the pure protein may indicate if this is an intra- or inter-molecular event.

## Discussion

Here we provide a rapid and robust procedure to purify recombinant mono-ubiquitinated proteins using the Avi-ubiquitin in a biochemical system. The single-step nature of affinity purification followed by protease-mediated elution of the modified proteins make this tech-nique amenable for use in the study of any ubiquitinated protein of research interest. We focused on the FA-BRCA pathway and were able to purify mono-ubiquitinated FANCI: FANCD2 complex, PCNA trimer and nucleosomes which can now be used as an reagent to answer long-standing questions in the field.

One of the challenges of researching ubiquitination is the low abundance of ubiquitinated proteins due to rapid turnover in cells. Previous protocols for purifying ubiquitinated proteins involve engineering a tandem ubiquitin-binding domain of interest [42] or immunoprecipita-tion of ubiquitinated peptides using anti-ubiquitin [43], however these methods often contain a mixture of poly- and mono-ubiquitination proteins. Moreover, enrichment of ubiquitinated peptides and mass spectrometry-based approach are required for protein identification, thus hindering identification of low abundance ubiquitinated protein complexes. Using our Avi-ubiquitin system, it may be possible to complement mass spectrometry-based approach and identify rare and mono-ubiquitinated proteins that are low abundance in cells, or identify new ubiquitin binding partners that could not be detected previously. A similar approach to Avi-ubiquitin utilized a GST-ubiquitin construct to purify ubiquitinated FANCI and FANCD2 monomeric proteins *in vitro* [24], however GST (26 kDa) [44] is a magnitude larger than ubiquitin (8 kDa) [45] and may not be suitable to purify smaller ubiquitinable proteins such as nucleosomes. To the best of our knowledge, this is the first study to purify natively mono-ubiquitinated nucleosome that was ubiquitinated by BRCA1:BARD1. This method

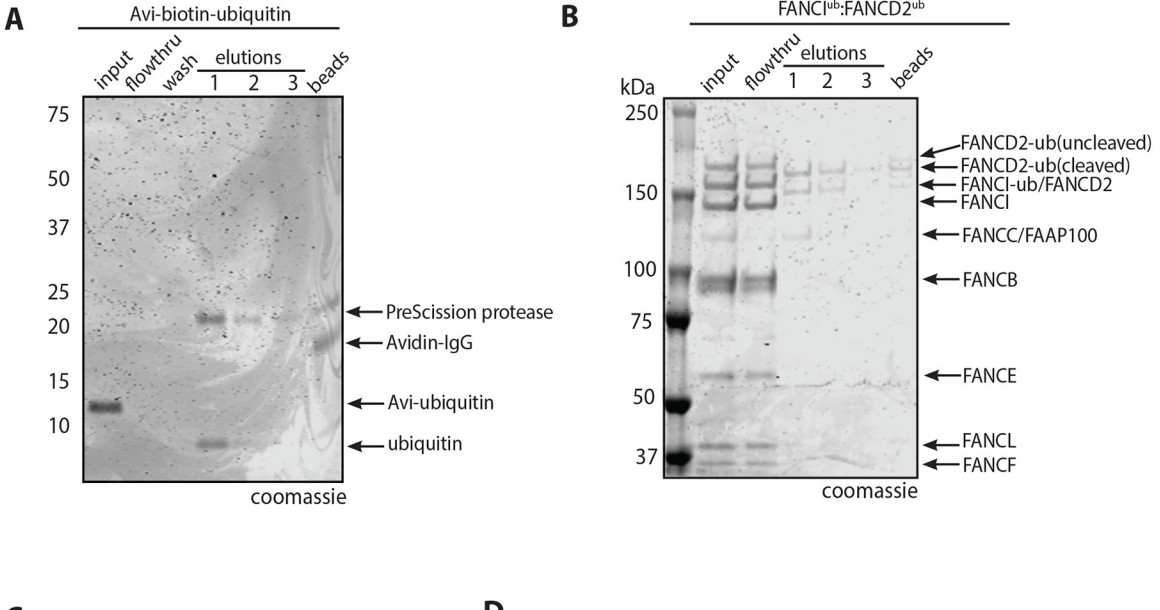

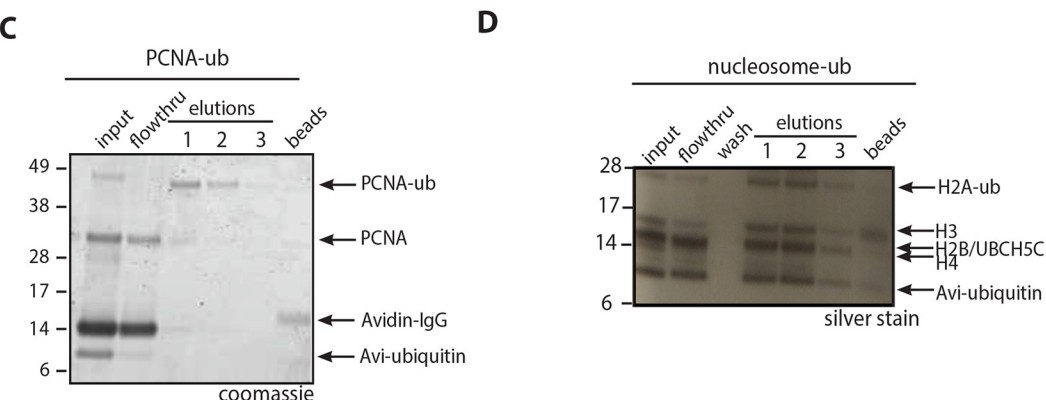

**Fig 3. Purification of mono-ubiquitinated proteins using the 3C protease enzyme. (A)** Coomassie of 4–12% Bis-Tris gel run in 1X MES buffer showing the purification Avi-ubiquitin. **(B)** Coomassie of 3–8% Tris-acetate gel showing the purification of mono-ubiquitinated FANCI:FANCD2 complex. **(C)** Coomassie stained SDS-PAGE gel showing purification of mono-ubiquitinated PCNA. **(D)** Silver stained SDS-PAGE gel showing purification of mono-ubiquitinated nucleosomes. Experiments in **(A-D)** were performed in triplicate with similar results.

provides a rapid and novel approach to purify mono-ubiquitinated proteins, and could potentially be applied to other poly-ubiquitination proteins.

The advantage of our protocol is that it is entirely reconstituted from purified components (ubiquitin, E3 ligase and DNA), which greatly simplifies data interpretation and allows more detailed investigation of the stepwise molecular events. Previous methods of purifying ubiquitinated PCNA utilized gel filtration to isolate ubiquitinated PCNA from the reconstituted reactions [18, 19], however this approach requires laborious optimization of the PCNA ubiquitination reaction and large amounts of ubiquitination reagents. Using our system, we were able to purify ubiquitinated PCNA away from unmodified substrates via a simple affinity chromatography step. Using Avi-ubiquitin, we were able to generate native dually-monoubiquitinated FANCI:FANCD2, which has been used as a reagent to study the FA-BRCA pathway [46].

One limitation of the system we describe is the need to extensively understand the biochemical mechanism of substrate monoubiquitination. For example, the generation of ubiquitinated FANCI-FANCD2 requires the reconstitution of a 6-protein E3 ligase complex.

Engineering of the UBE2T enzyme to increase its specificity has recently been demonstrated as a way to overcome this requirement [47] however this requires a tailored approach for each target. We observed very little non-specific monoubiquitination in our reactions, however for some E2:E3 pairs this may be an issue to be overcome. In all our experiments, Avi-ubiquitin has behaved identically to unlabeled ubiquitin, however this may also be different if alternative E2:E3 pairs and substrates are used. A further limitation, related to purification of the ubiquitinated proteins, is the requirement that the N-terminus of ubiquitin be solvent exposed in the monoubiquitinated substrate. Ubiquitin that is protected by binding to an additional factor or protein domain will not be accessible to Streptavidin for capture. Importantly, the Avi-ubiquitin is biotinylated *in vivo* in *E.coli* and then purified via a HIS-tag. As such, so no other biotin-modified proteins should participate in the reaction. This limits the contaminant issues that come from *in vitro* biotinylation of ubiquitin or ubiquitinated proteins for Streptavidin purification.

The Avi-ubiquitin purification protocol presented here is widely applicable and represents an immediate step towards direct purification of native protein complexes *in vitro* or in cells. Our approach takes advantage of the strong affinity between avidin and biotin, which opens the door for purifying endogeneous ubiquitinated protein complexes from a number of organisms that have not yet been explored, enabling capturing of multiple natively ubiquitinated proteins or binding partners (for example different ubiquitinated substrates in the Proteasome degradation system [48] or ubiquitinated proteins in different cell cycle stages) [49, 50], and during various stages of diseases (cancer, autoimmune or cardiovascular diseases).

## Supporting information

**S1 Fig. Purification and mono-ubiquitination of human FANCI:FANCD2 complex. (A)** Coomassie stained 3–8% Tris-acetate gel of Flag- and Talon-affinity purification of human FANCI:FANCD2 complex, and Western blot of Talon-purified FANCI:FANCD2 complex. **(B)** Coomassie stained SDS-PAGE gel revealing that human FANCI runs higher than *Xenopus* FANCI in the FANCI:FANCD2 complex. **(C)** Mono-ubiquitination of human and *Xenopus* FANCI:FANCD2 complex in a reaction containing recombinant FA core complex proteins at 25˚C for 90 min.
(PDF)

**S2 Fig. Biochemical reconstitution of mono-ubiquitinated nucleosomes. (A)** Coomassie stained SDS-PAGE showing reconstitution of mono-ubiquitinated nucleosomes. Lane 1 shows 147 bp dsDNA based on Widom 601 sequence in the absence of nucleosome (Widom 147), lane 2 shows mono-ubiquitination reaction containing nucleosomes without ATP (nucleosomes) and lane 3 shows mono-ubiquitination reaction containing nucleosomes, BRCA1$^{\Delta exon11}$:BARD1 complex, Avi-biotin-ubiquitin, UBE1, UBCH5C and ATP (ubiquitinated nucleosomes). **(B)** 6% native PAGE gel containing the same reactions in (A) stained with SYBR gold nucleic acid stain showing reconstitution of nucleosomes (lanes 2–3). **(C)** Western blot of the 6% P AGE from (B) probed against Streptavidin antibody showing that the reconstituted nucleosomes were ubiquitinated (lane 3).
(PDF)

**S3 Fig. Mass spectrometry results of mono-ubiquitinated FANCI, FANCD2,PCNA, BRCA1 and BARD1. (A-C)** A gel band containing ubiquitinated FANCI, FANCD2 and PCNA were processed and in-gel digested with trypsin and analyzed by mass spectrometry. The tandem mass (M S/M S) spectrum of mono-ubiquitinated peptides were derived by collision-induced dissociation of the (M+H)+ precursor, m/z as indicated. Fragment ions in the

spectrum represent mainly single-event preferential cleavage of the peptide bonds resulting in the sequence information recorded simultaneously from both the N- and C-termini (b- and y-type ions, respectively) of the peptide. Data analysis was performed using M ascot (35) and the data was searched against the Uniprot database (downloaded 0606/2013). Searches were conducted with trypsin, the precursor ion tolerance was set to 10 ppm and the fragment ion tolerance was set to 0.2 Da. Variable modifications include oxidation on methionine (+15.995 Da), carbamidomethyl (+57.021 Da) on cysteine, and ubiquitination on glycine/glycine (+114.043 Da). The maximum missed cleavages were set to 2. All search results were evaluated by MudPIT scoring for false discovery rate (FDR) evaluation of the identified peptides (36) and peptides identifications were filtered to a FDR of 5%. Mass spectra verified that the purified FANCI is mono-ubiquitinated at lysine 523 **(A)**, FANCD2 is mono-ubiquitinated at lysine 561 **(B)**, and PCNA is mono-ubiquitinated at lysine 167 **(C). (D)** Summary of mass spectrometry results of mono-ubiquitinated BRCA1:BARD1.
(PDF)

**S4 Fig. Source data gels.** Original source images for all data obtained by electrophoretic separation: Coomassie stained SDS-PAGE and western blots.
(PDF)

## Acknowledgments

We thank Rachel Klevit and David Waugh for reagents. Mass spectrometry was performed at the Bio21 Institute Mass Spectrometry and Proteomics facility.

## Author Contributions

**Conceptualization:** Winnie Tan, Wayne Crismani, Rohan Bythell-Douglas, Andrew J. Deans.

**Formal analysis:** Winnie Tan, Rohan Bythell-Douglas, Andrew J. Deans.

**Funding acquisition:** Rohan Bythell-Douglas, Andrew J. Deans.

**Investigation:** Winnie Tan, Vincent J. Murphy, Aude Charron, Sylvie van Twest, Wayne Crismani, Andrew J. Deans.

**Methodology:** Winnie Tan, Vincent J. Murphy, Aude Charron, Sylvie van Twest, Rohan Bythell-Douglas, Andrew J. Deans.

**Project administration:** Rohan Bythell-Douglas, Andrew J. Deans.

**Resources:** Sylvie van Twest, Michael Sharp, Angelos Constantinou.

**Supervision:** Vincent J. Murphy, Michael W. Parker, Wayne Crismani, Rohan Bythell-Douglas, Andrew J. Deans.

**Validation:** Winnie Tan.

**Visualization:** Winnie Tan, Andrew J. Deans.

**Writing – original draft:** Winnie Tan, Andrew J. Deans.

**Writing – review & editing:** Winnie Tan, Aude Charron, Angelos Constantinou, Michael W. Parker, Wayne Crismani, Rohan Bythell-Douglas, Andrew J. Deans.

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
