## [Decision Letter · Decision Letter 0]

27 Dec 2019

PONE-D-19-34194

Preparation and purification of mono-ubiquitinated proteins using Avi-tagged ubiquitin

PLOS ONE

Dear Dr Deans,

Thank you for submitting your manuscript to PLOS ONE. After careful consideration, we feel that it has merit but does not fully meet PLOS ONE’s publication criteria as it currently stands. Therefore, we invite you to submit a revised version of the manuscript that addresses the points raised during the review process.

Both reviewers have identified minor but important points that should be addressed. Please attend to all the suggestions requested. 

We would appreciate receiving your revised manuscript by Feb 10 2020 11:59PM. To enhance the reproducibility of your results, we recommend that if applicable you deposit your laboratory protocols in protocols.io, where a protocol can be assigned its own identifier (DOI) such that it can be cited independently in the future. For instructions see: http://journals.plos.org/plosone/s/submission-guidelines#loc-laboratory-protocols

We look forward to receiving your revised manuscript.

Kind regards,

Robert W Sobol, PhD

Academic Editor

PLOS ONE

Journal Requirements:

5. Please include captions for your Supporting Information files at the end of your manuscript, and update any in-text citations to match accordingly. Please see our Supporting Information guidelines for more information: http://journals.plos.org/plosone/s/supporting-information

Reviewers' comments:

Reviewer's Responses to Questions

**Comments to the Author**

1. Is the manuscript technically sound, and do the data support the conclusions?

Reviewer #1: Yes

Reviewer #2: Yes

2. Has the statistical analysis been performed appropriately and rigorously? 

Reviewer #1: N/A

Reviewer #2: I Don't Know

3. Have the authors made all data underlying the findings in their manuscript fully available?

Reviewer #1: Yes

Reviewer #2: Yes

4. Is the manuscript presented in an intelligible fashion and written in standard English?

Reviewer #1: Yes

Reviewer #2: Yes

5. Review Comments to the Author

Reviewer #1: In this manuscript, the authors designed ubiquitin fusion protein that can be biotinylated by BirA and can be conjugated by E3 ligases to substrates. After biotin affinity purification of modified targets, mono-ubiquitinated proteins can be purified and used for other study. This method may provide a simple strategy to study the role of ubiquitinated protein in vitro. The data support the conclusion.

Figure 1C: need to label the lane with number. This figure shows that the elution is not very efficient, have you calculated which percent of product is not eluted in the experiment?

Figure 2A: need to label the lane with number.

Figure 2B: It is very difficult to differ FANCD2-ub and FANCD2 in the image, how the percentage was calculated?

Supple Fig 3: please indicates which line is the ubiquitylated ions in the graph.

Fig 2B and 2D: please provide the images of full blot.

Reviewer #2: The authors present a novel in vitro method for the identification of natively-mono-ubiquitinated substrates using Biotin affinity purification and apply it to the Fanconi anemia (FA)-BRCA pathway of DNA repair.

A few minor comments that should be addressed:

1) No statistics have been performed and the number of replicates remain unknown.

2) The text in the body of the manuscript for Figure 1C does not correspond with the Figure. Please correct.

3) A brief summary of the potential limitations of the Avi-ubiquitin purification protocol and the application presented in this manuscript (for example, any non-specific biotinlyation and subsequent non-specific mono-ubiquitination observed) will be helpful for scientists replicating this procedure.

6. PLOS authors have the option to publish the peer review history of their article (what does this mean?). If published, this will include your full peer review and any attached files.

Reviewer #1: Yes: Qingming Fang

Reviewer #2: No

---

## [Author Response · Author response to Decision Letter 0]

19 Jan 2020

We thank the reviewers for their comments and constructive feedback. Specific points are addressed below and throughout the revised manuscript.

Reviewer #1: In this manuscript, the authors designed ubiquitin fusion protein that can be biotinylated by BirA and can be conjugated by E3 ligases to substrates. After biotin affinity purification of modified targets, mono-ubiquitinated proteins can be purified and used for other study. This method may provide a simple strategy to study the role of ubiquitinated protein in vitro. The data support the conclusion.

Figure 1C: need to label the lane with number. This figure shows that the elution is not very efficient, have you calculated which percent of product is not eluted in the experiment?

*Figure 1C is now labelled with lane number as suggested. The percentage of Avi-ubiquitin eluted is only ~50% by using 5 mM biotin, however cleavage of ubiquitin using 3C protease resulted in >95% elution of ubiquitin as indicated in Figure 1C and Figure 3A. The percentage of Avi-ubiquitin eluted is now included in the updated manuscript as requested (lines 277-281).

Figure 2A: need to label the lane with number.

*Lane number is now included in Figure 2A.

Figure 2B: It is very difficult to differ FANCD2-ub and FANCD2 in the image, how the percentage was calculated?

*The top panel of Figure 2B is enlarged to show bands corresponding to FANCD2-ub and FANCD2. The percentage of FANCD2-ub is calculated by quantifying the intensity of FANCD2 and FANCD2-ub bands using ImageJ, and divide the intensity of FANCD2-ub by total intensity of FANCD2.

Supple Fig 3: please indicates which line is the ubiquitylated ions in the graph.

*The spectra shown in this figure correspond to the ms/ms of the b+ and y+ ions of a single peptide. Various b+ and y+ ions correspond to different fragments of the ubiquitinated peptide and together accurately identify the ubiquitinated residues indicated.

Fig 2B and 2D: please provide the images of full blot.

*Images of full blot for Fig 2B and 2D are now included in Supp Fig 4.

Reviewer #2: The authors present a novel in vitro method for the identification of natively-mono-ubiquitinated substrates using Biotin affinity purification and apply it to the Fanconi anemia (FA)-BRCA pathway of DNA repair.

A few minor comments that should be addressed:

1) No statistics have been performed and the number of replicates remain unknown.

*The number of replicates (n=3) is now added in all Figure legends as required. Similar results were achieved in triplicates.

2) The text in the body of the manuscript for Figure 1C does not correspond with the Figure. Please correct.

*The text in the body of manuscript (line 259-286) is amended to correspond to the figure 1C.

3) A brief summary of the potential limitations of the Avi-ubiquitin purification protocol and the application presented in this manuscript (for example, any non-specific biotinlyation and subsequent non-specific mono-ubiquitination observed) will be helpful for scientists replicating this procedure.

*A brief summary is added in the manuscript body from line 367 onwards.

---

## [Decision Letter · Decision Letter 1]

29 Jan 2020

Preparation and purification of mono-ubiquitinated proteins using Avi-tagged ubiquitin

PONE-D-19-34194R1

Dear Dr. Deans,

We are pleased to inform you that your manuscript has been judged scientifically suitable for publication and will be formally accepted for publication once it complies with all outstanding technical requirements.

With kind regards,

Robert W Sobol, PhD

Academic Editor

PLOS ONE

Additional Editor Comments (optional):

Reviewers' comments:

Reviewer's Responses to Questions

**Comments to the Author**

1. If the authors have adequately addressed your comments raised in a previous round of review and you feel that this manuscript is now acceptable for publication, you may indicate that here to bypass the “Comments to the Author” section, enter your conflict of interest statement in the “Confidential to Editor” section, and submit your "Accept" recommendation.

Reviewer #1: All comments have been addressed

Reviewer #2: All comments have been addressed

2. Is the manuscript technically sound, and do the data support the conclusions?

Reviewer #1: Yes

Reviewer #2: (No Response)

3. Has the statistical analysis been performed appropriately and rigorously? 

Reviewer #1: Yes

Reviewer #2: (No Response)

4. Have the authors made all data underlying the findings in their manuscript fully available?

Reviewer #1: Yes

Reviewer #2: (No Response)

5. Is the manuscript presented in an intelligible fashion and written in standard English?

Reviewer #1: Yes

Reviewer #2: (No Response)

6. Review Comments to the Author

Reviewer #1: (No Response)

Reviewer #2: (No Response)

7. PLOS authors have the option to publish the peer review history of their article (what does this mean?). If published, this will include your full peer review and any attached files.

Reviewer #1: No

Reviewer #2: No

---

## [Editor Report · Acceptance letter]

10 Feb 2020

PONE-D-19-34194R1 

Preparation and purification of mono-ubiquitinated proteins using Avi-tagged ubiquitin 

Dear Dr. Deans:

I am pleased to inform you that your manuscript has been deemed suitable for publication in PLOS ONE. Congratulations! Your manuscript is now with our production department. 

With kind regards,

on behalf of

Dr. Robert W Sobol 

Academic Editor

PLOS ONE